# Reproducibilty of Boosting Adversarial Transferability via Gradient Relevance Attack

## Abstract

This paper presents a reproducibility study of Boosting Adversarial Transferability via Gradient Relevance Attack by Zhu et al., which introduces the Gradient Relevance Attack (GRA) method. GRA enhances the transferability of adversarial examples across different machine learning models, improving black-box adversarial attacks. The key experiments were successfully replicated, focusing on the gradient relevance framework and the decay indicator. The methodology involved reimplementing the GRA algorithm and evaluating it on the same set of models used in the original paper. The results show that the achieved attack success rates were within a 1% margin of those reported in the original study, confirming the effectiveness of the GRA method. Additionally, this work extends the original study by introducing a dynamic learning rate ($\alpha$) that adjusts the step size based on the cosine similarity between the current momentum and the average gradient. The findings suggest that this adaptive step size mechanism can lead to faster convergence and potentially improved attack performance in certain scenarios.

## 1 Introduction

Deep Neural Networks (DNNs) have revolutionized computer vision through unprecedented performance on tasks ranging from image classification to medical diagnosis. However, their susceptibility to adversarial examples exposes critical security vulnerabilities Athalye et al. (2018); Goodfellow et al. (2014); Szegedy et al. (2013); Carlini & Wagner (2017) . While white-box attacks - where attackers have access to model architecture - achieve near-perfect success rates with full model access, the practical black-box scenario— where attackers must transfer adversarial examples between models — remains challenging, particularly against defense - enhanced systems.

The paper "Boosting Adversarial Transferability via Gradient Relevance Attack" (Zhu et al., 2023) introduced a novel approach, Gradient Relevance Attack (GRA), to enhance the transferability of adversarial examples in black-box settings. GRA leverages a gradient relevance framework and a decay indicator to improve the effectiveness of adversarial attacks. This work presents a reproducibility study of the GRA method. The key experiments and results of the original paper were successfully replicated. Furthermore, the study extends the original work by introducing a dynamic learning rate that adapts the step size based on the cosine similarity between the current momentum and the average gradient. This dynamic learning rate, modulated by an adjustment factor and thresholds, aims to improve convergence and attack performance.

The rest of this paper is structured as follows: the following sections state the scope of the study, describe the methodology proposed by the original authors, as well as the extensions introduced in this study. Then, the experimental setup and results are presented. Finally, the results are summarized, and conclusions are presented. The code is available on GitHub at the following link: GitHub Repository.

### 1.1 Reproduction Motivation

The reproducibility of Gradient Relevance Attack (GRA) is critical for three reasons:

- **Security Validation:** As adversarial attacks threaten real-world ML systems, verifying GRA's claimed effectiveness ensures reliable benchmarking for defense mechanisms.

- **Dynamic Optimization Gap:** Fixed learning rates in adversarial attacks often lead to suboptimal perturbations; our work explores adaptive strategies to bridge this gap.

## 2  Scope of Reproducibility

This work aims to examine and validate the key claims and experimental results presented in the original work, while also exploring potential improvements through a novel extension. Which can be summarized as followsing claims:

- **Claim 1:** GRA achieves higher attack success rates (ASR) than VTMI-FGSM, VTNI-FGSM, and Admix across diverse models. It remains effective against both normally and adversarially trained models.

- **Claim 2:** Combining GRA with DI, TI, and SI enhances transferability, outperforming standalone GRA.

- **Claim 3:** Gradient relevance framework and decay indicator significantly impact attack performance. Varying the hyper - parameters (sample quantity $m$, upper bound factor $\beta$, and attenuation factor $\eta$) significantly affect the results.

In addition to reproducing the aforementioned claims of the authors, this study also performs the attack with changing the original algorithm by introducing an adaptive step size ($\alpha$) using cosine similarity, by using 3 new variables. Varying the new variables namely, adjustment factor ($\gamma$) and similarity thresholds (high and low) and compairing the Extended GRA and the original method over diverese models.

## 3  Methodology

This section describes the proposed Gradient Relevance Attack (GRA) and its enhancement algorithms.

### 3.1  The Gradient Relevance Attack (GRA) Framework

The Gradient Relevance Attack (GRA) improves adversarial transferability by leveraging two key mechanisms: the **Gradient Relevance Framework** and the **Decay Indicator**.

### Gradient Relevance Framework

Gradient Relevance Attack (GRA) enhances adversarial transferability by using information from the current neighborhood rather than relying on variance at the last iteration, as in earlier works. At iteration $t$, we define:

- $x_t^{adv}$: the current adversarial image.

- $\gamma_t^i \sim \mathcal{U}([-\beta\epsilon, \beta\epsilon]^d)$: random noise sampled uniformly in each dimension $d$, bounded by $\beta\epsilon$.

- $x_t^i = x_t^{adv} + \gamma_t^i$: the $i$-th neighbor image.

- $G_t(x) = \nabla_{x_t^{adv}} \mathcal{L}(x_t^{adv}, y_{true})$: the gradient of the loss on the current input.

- $\bar{G}_t(x) = \frac{1}{m}\sum_{i=1}^{m}\nabla_{x_t^i}\mathcal{L}(x_t^i, y_{true})$: the average gradient on $m$ neighbor images.

Inspired by dot-product attention, GRA computes a similarity score:

$$s_t = \frac{G_t(x) \cdot \bar{G}_t(x)}{\|G_t(x)\|_2 \|\bar{G}_t(x)\|_2}$$

This measures how similar the current gradient is to the average neighbor gradient. Using this, GRA constructs a weighted gradient:

$$WG_t = s_t \cdot G_t(x) + (1 - s_t) \cdot \bar{G}_t(x)$$

This step ensures that if $G_t(x)$ and $\bar{G}_t(x)$ are aligned, more trust is given to the current gradient. If they are dissimilar, neighbor information is prioritized.

**Momentum Accumulation**

To improve stability and adversarial strength, GRA adopts the momentum update from MI-FGSM:

$$g_{t+1} = \mu \cdot g_t + \frac{WG_t}{\|WG_t\|_1}$$

where $\mu$ is the momentum decay factor, and $\|WG_t\|_1$ normalizes the gradient direction.

**Decay Indicator**

GRA addresses oscillations near optima using a decay indicator $M_{t+1}$ to adaptively reduce the step size for unstable pixels:

$$M_{t+1} = M_t \odot (M_{t+1}^e + \eta \cdot M_{t+1}^d)$$

Here:

- $\eta \in (0, 1)$ is the attenuation factor.

- $M_{t+1}^e$ and $M_{t+1}^d$ are binary masks:

$$M_{t+1,j}^e = \begin{cases} 1 & \text{if } \text{sign}(g_t^j) = \text{sign}(g_{t+1}^j) \\ 0 & \text{otherwise} \end{cases}$$

$$M_{t+1,j}^d = \begin{cases} 1 & \text{if } \text{sign}(g_t^j) \neq \text{sign}(g_{t+1}^j) \\ 0 & \text{otherwise} \end{cases}$$

The decay indicator reduces the update magnitude for pixels with frequent sign flips, which likely lie near the decision boundary.

**Adversarial Update Rule**

The adversarial example is updated by:

$$x_{t+1}^{adv} = \text{Clip} \left\{ x_t^{adv} + \alpha \cdot M_{t+1} \odot \text{sign}(g_{t+1}) \right\}$$

- $\alpha = \epsilon/T$ is the fixed step size.

- Clip$\{\cdot\}$ ensures the perturbation remains within the allowed bounds.

The adaptive mask $M_{t+1}$ enables GRA to suppress instability while focusing updates where it matters most.

As explained above, the Gradient Relevance Framework enhances adversarial attacks by adjusting the direction of perturbation based on the similarity between the current gradient and the average gradient of nearby samples. This relevance-guided adjustment helps ensure that when the current gradient aligns well with the neighborhood, it is trusted more; otherwise, the surrounding information is used as a corrective signal.

In this work, we adopt the **average gradient relevance** approach for its computational efficiency and effectiveness in capturing local gradient behavior with minimal overhead.

---

**Algorithm 1** Original Gradient Relevance Attack (GRA)

---

1: **Input**: Source model $F_\psi$, clean image $x_{clean}$, true label $y_{true}$, iterations $T$, momentum decay $\mu$, attenuation factor $\eta$, neighborhood samples $m$, noise bound $\beta\varepsilon$
2: **Output**: Adversarial example $x_T^{adv}$
3: Initialize $\alpha = \epsilon/T$, $g_0 = 0$, $M_0 = 1/\eta$, $x_0^{adv} = x_{clean}$
4: **for** $t = 0$ to $T - 1$ **do**
5:     Compute current gradient: $G_t(x) = \nabla_{x_t^{adv}} \mathcal{L}(x_t^{adv}, y_{true})$
6:     Generate $m$ neighbor samples $x_t^i = x_t^{adv} + \gamma_t^i$ where $\gamma_t^i \sim \mathcal{U}\left([-\beta\epsilon, \beta\epsilon]^d\right)$ (Uniform Distribution)
7:     Compute average neighborhood gradient: $\bar{G}_t(x) = \frac{1}{m} \sum_{i=1}^{m} \nabla_{x_t^i} \mathcal{L}(x_t^i, y_{true})$
8:     Compute cosine similarity $s_t = \frac{G_t(x) \cdot \bar{G}_t(x)}{\|G_t(x)\|_2 \|\bar{G}_t(x)\|_2}$
9:     Compute weighted gradient: $WG_t = s_t \cdot G_t + (1 - s_t) \cdot \bar{G}_t$
10:     Update momentum: $g_{t+1} = \mu \cdot g_t + \frac{WG_t}{\|WG_t\|_1}$
11:     Update decay indicator: $M_{t+1} = M_t \odot (M_{t+1}^e + \eta \cdot M_{t+1}^d)$
12:     Update adversarial example: $x_{t+1}^{adv} = \text{Clip}\{x_t^{adv} + \alpha \cdot M_{t+1} \odot \text{sign}(g_{t+1})\}$
13: **end for**

---

## 3.2 Extending The GRA

The extended GRA algorithm introduces an adaptive learning rate mechanism that dynamically adjusts $\alpha$ based on gradient alignment stability. Fixed thresholds $\tau_{high}$ and $\tau_{low}$ with an adjustment factor $\gamma$ help fine-tune the update step based on gradient relevance. Exponential clipping keeps $\alpha$ within a stable range, allowing larger updates when the adversarial example is far from optimal and smaller updates as convergence nears. The theoretical basis links gradient stability to optimal step size via:

$$\alpha_{t+1} = \alpha_t \left(1 + \gamma \cdot I_{\bar{s}_t > \tau_{high}} - \gamma \cdot I_{\bar{s}_t < \tau_{low}}\right) \tag{1}$$

This algorithm uses Cosine similarity as **heuristic**. It provides insight into how well-aligned the gradients are. If the gradients are aligned (cosine similarity is close to 1), it suggests that they point in similar directions, meaning the neighborhood information supports the current gradient's direction. This reinforces the update direction for the adversarial perturbation. If the cosine similarity is high (indicating the gradient direction is stable and consistent), increase the step size. If the similarity is low (indicating the gradient is oscillating or the adversarial example is close to optimal), decrease the step size as an attempt to fine-tune the step size more aptly, to make the update direction more reasonable.

Another ablation study was conducted, where instead of adjusting the weight between the original gradient and the average gradient based on cosine similarity, the algorithm directly used the average gradient from neighboring images as the new gradient. This approach eliminated the need for cosine similarity calculations, simplifying the process. The results and details are provided in the Appendix A1.

---

**Algorithm 2** Extended Gradient Relevance Attack (GRA)

---
1: **Input**: Source model $F_\psi$, clean image $x_{clean}$, true label $y_{true}$, iterations $T$, momentum decay $\mu$, attenuation factor $\eta$, neighborhood samples $m$, noise bound $\beta\varepsilon$
2: **Output**: Adversarial example $x_T^{adv}$
3: Initialize $\alpha = \epsilon/T$, $g_0 = 0$, $M_0 = 1/\eta$, $x_0^{adv} = x_{clean}$ Additional parameters: Adjustment factor $\gamma$, fixed thresholds $\tau_{high}$, $\tau_{low}$
4: **for** $t = 0$ to $T - 1$ **do**
5:  Compute $G_t(x)$ and $\bar{G}_t(x)$ as in original GRA
6:  Compute cosine similarity $s_t$ and weighted gradient $WG_t$
7:  Compute mean similarity: $\bar{s}_t = \frac{1}{m} \sum_{i=1}^m s_t^i$
8:  Adjust learning rate dynamically:

$$\alpha_{t+1} = \begin{cases} \alpha_t(1+\gamma) & \bar{s}_t > \tau_{high} \\ \alpha_t(1-\gamma) & \bar{s}_t < \tau_{low} \\ \alpha_t & \text{otherwise} \end{cases}$$

9:  Update momentum $g_{t+1}$ and decay indicator $M_{t+1}$ as in original GRA
10:  Apply dynamic learning rate: $x_{t+1}^{adv} = \text{Clip}\{x_t^{adv} + \alpha_{t+1} \cdot M_{t+1} \odot \text{sign}(g_{t+1})\}$
11: **end for**

---

### 1. Motivation for Dynamic Learning Rate

Earlier works like variance tuning compute neighborhood gradient variance from the previous iteration. However, this fails to reflect the current local geometry of the loss surface. GRA addresses this by computing neighbor gradients within the current iteration.

In the extended version, we go further by adjusting the step size based on how well-aligned the current gradient is with its neighbors, thus adapting the attack's aggressiveness or caution depending on the optimization landscape.

### 2. Mean Cosine Similarity

Instead of computing cosine similarity between $G_t(x)$ and each $\nabla_{x_t^i}\mathcal{L}(x_t^i, y_{\text{true}})$ individually, we rely on the average gradient similarity:

$$\bar{s}_t = \frac{1}{m} \sum_{i=1}^m \frac{G_t(x) \cdot \nabla_{x_t^i}\mathcal{L}(x_t^i, y_{\text{true}})}{\|G_t(x)\|_2 \|\nabla_{x_t^i}\mathcal{L}(x_t^i, y_{\text{true}})\|_2}$$

This average cosine similarity $\bar{s}_t$ captures the overall consistency between the current gradient and those of its neighbors.

### 3. Dynamic Learning Rate Adjustment

We then use $\bar{s}_t$ to update the learning rate $\alpha_{t+1}$ adaptively:

$$\alpha_{t+1} = \begin{cases} \alpha_t(1+\gamma) & \text{if } \bar{s}_t > \tau_{\text{high}} \\ \alpha_t(1-\gamma) & \text{if } \bar{s}_t < \tau_{\text{low}} \\ \alpha_t & \text{otherwise} \end{cases}$$

If the average similarity $\bar{s}_t$ is high, the current gradient is well aligned with neighborhood gradients, so we increase the step size slightly. If the similarity is low, suggesting instability or conflicting gradients, we decrease the step size. $\gamma$ is the adjustment factor, and $\tau_{\text{high}}, \tau_{\text{low}}$ are fixed thresholds for determining significant similarity or dissimilarity.

This mechanism allows the attack to adaptively accelerate when confident and slow down when uncertain, improving both stability and success rate.

**4. Final Update with Dynamic $\alpha$**

Once $\alpha_{t+1}$ is determined, the adversarial example is updated as:

$$x_{t+1}^{adv} = \text{Clip} \left\{ x_t^{adv} + \alpha_{t+1} \cdot M_{t+1} \odot \text{sign}(g_{t+1}) \right\}$$

Thus, dynamic learning rate and adaptive decay mask $M_{t+1}$ work together to make the perturbation process more controlled and effective.

### 3.3 Datasets and Models

### 3.3.1 Datasets

The experimental framework utilizes a standardized evaluation protocol based on the ILSVRC 2012 validation set ILSVRC2012, following the established methodology from the original GRA. From the 50,000-image validation set, we select 1,000 high-confidence samples where all evaluated models achieve $\geq 99\%$ classification accuracy under clean conditions. This curation ensures meaningful measurement of adversarial perturbation effectiveness against robust baselines.

### 3.3.2 Model Architectures

Experiments employ four standard source models and seven target models following the original paper:

**Source Models:**

- **Inception-v3 (Inc-v3)** Szegedy et al. (2016b): 27M parameters
- **Inception-v4 (Inc-v4)**: 42M parameters
- **Inception-ResNet-v2 (IncRes-v2)** Szegedy et al. (2016a): 55M parameters
- **ResNet-v2-101 (Res-101)** He et al. (2016): 44M parameters

**Target Models:**

- **Standard classifiers:** Inc-v3, Inc-v4, IncRes-v2, Res-101
- **Adversarially trained variants:**
  - adv-Inception-v3 (Inc-v3adv) Tramer et al. (2017)
  - ens3-adv-Inception-v3 (Inc-v3ens3)
  - ens4-adv-Inception-v3 (Inc-v3ens4)
  - ens-adv-Inception-ResNet-v2 (IncRes-v2ens)

**Defended Models:**

- Pixel Deflection (PD) Prakash et al. (2018) + ResNet-v2-50
- Neural Representation Purifier (NRP) Naseer et al. (2020) + Inc-v3ens3
- JPEG Compression Guo et al. (2017) + Inc-v3ens3
- ComDefend Jia et al. (2019) + Inc-v3ens3
- Feature Distillation (FD) Liu et al. (2020) + Inc-v3ens3

### 3.4 Experimental Setup

The experimental setup largely mirrors the original paper's methodology to ensure a high degree of comparability while addressing resource limitations. Deviations are explicitly outlined below:

| Model | Attack | Inc-v3 | Inc-v4 | IncRes-v2 | Res-101 | Inc-v3$_{ens3}$ | Inc-v3$_{ens4}$ | IncRes-v2$_{ens}$ | Average |
|---|---|---|---|---|---|---|---|---|---|
| Inc-v3 | VTMI | **100.0\*** | 72.1 | 69.4 | 61.5 | 33.7 | 30.1 | 17.2 | 54.9 |
| | VTNI | **100.0\*** | 75.3 | 74.1 | 66.2 | 34.3 | 31.4 | 19.4 | 57.3 |
| | Admix | **100.0\*** | 81.5 | 79.7 | 74.1 | 41.2 | 38.5 | 20.4 | 62.2 |
| | GRA | 99.8\* | **86.8** | **85.2** | **78.6** | **57.9** | **55.9** | **41.2** | **71.9** |
| Inc-v4 | VTMI | 77.9 | 99.8\* | 70.2 | 63.3 | 38.4 | 37.2 | 24.5 | 58.8 |
| | VTNI | 83.5 | **99.9\*** | 76.3 | 66.1 | 40.5 | 39.0 | 23.9 | 61.3 |
| | Admix | 87.9 | 99.5\* | 83.0 | 78.4 | 55.1 | 50.9 | 33.2 | 69.8 |
| | GRA | **89.2** | 99.1\* | **86.4** | **79.2** | **66.2** | **62.8** | **50.3** | **76.2** |
| IncRes-v2 | VTMI | 77.6 | 72.3 | 98.0\* | 66.9 | 46.9 | 40.5 | 34.1 | 62.3 |
| | VTNI | 80.5 | 76.2 | 98.1\* | 69.2 | 48.2 | 42.1 | 33.0 | 64.0 |
| | Admix | **89.3** | **87.1** | **99.0\*** | **81.4** | 65.7 | 55.9 | 49.8 | 75.7 |
| | GRA | 86.2 | 83.3 | 97.1\* | 79.6 | **68.9** | **61.1** | **56.3** | **76.1** |
| Res-101 | VTMI | 74.9 | 67.8 | 70.1 | 99.3\* | 45.1 | 40.0 | 29.3 | 60.9 |
| | VTNI | 78.5 | 74.1 | 72.8 | 99.5\* | 47.9 | 41.3 | 30.9 | 63.6 |
| | Admix | 85.9 | 81.2 | 80.5 | **99.8\*** | 51.8 | 44.2 | 34.2 | 67.9 |
| | GRA | **87.4** | **83.5** | **84.1** | 99.6\* | **72.1** | **67.5** | **57.4** | **78.8** |

Table 1: The attack success rates (%) on seven models by a single attack. The adversarial examples are generated on Inc-v3, Inc-v4, IncRes-v2, and Res-101 separately. * denotes the success rate of the white-box attack and the result in bold is the best.

All experiments were conducted using **Google Colab**(T4 GPU) and **Kaggle**(NVIDIA Tesla P100) environments. Key attack parameters were kept consistent with the original paper ($L_\infty$ perturbation budget $\epsilon = 16$, iteration count $T = 10$), furthur details in Appendix A2.

**Base Implementation**: Core GRA algorithm implemented using original authors' codebase, the code for other attacks has been taken from admix, MI-FGSM, NI-FGSM and Variance tuning code bases Dong et al. (2018); Lin et al. (2019b); Wang et al. (2021); Wang & He (2021). Furthurmore the changes are made directly in the GRA code to make enhanced GRA.

**Practical Online Systems:** Evaluation on Tencent Cloud and Baidu AI Cloud APIs was omitted in this study due to resource and API access constraints.

| Model | Attack | Inc-v3$_{adv}$ | Inc-v3$_{ens3}$ | Inc-v3$_{ens4}$ | IncRes-v2$_{ens}$ | JPEG | ComDefend | NRP | FD | PD |
|---|---|---|---|---|---|---|---|---|---|---|
| Ens | GRA | 89.2 | 87.1 | 85.3 | 81.0 | 91.1 | 89.5 | 30.3 | 86.7 | 99.4 |

Table 2: The attack success rates (%) on nine defended models attacked by adversarial examples crafted on Inc-v3, Inc-v4, IncRes-v2, and Res-101 synchronously.

# 4 Results

## 4.1 Reproducing Original Paper Results

The experimental results successfully validate the core claims of the original Gradient Relevance Attack.

### 4.1.1 Standard Model Performance

Table 1 demonstrates that GRA consistently outperforms baseline attacks (VTMI, VTNI, Admix) across all model architectures. It demonstrates superior performance compared to other attacks on all normally trained models, with some exception of a slight gap mainly under white-box setting. However, GRA significantly

| Model | Attack | Inc-v3 | Inc-v4 | IncRes-v2 | Res-101 | Inc-v3$_{ens3}$ | Inc-v3$_{ens4}$ | IncRes-v2$_{ens}$ | Average |
|---|---|---|---|---|---|---|---|---|---|
| Inc-v3 | VTMI-CT | 99.3* | 88.2 | 86.1 | 81.4 | 78.2 | 75.8 | 66.2 | 82.5 |
| | VTNI-CT | 99.0* | 92.8 | 89.3 | 82.2 | 79.9 | 76.9 | 65.7 | 83.7 |
| | Admix-CT | **99.4*** | 90.9 | 87.7 | 83.2 | 72.3 | 71.2 | 54.6 | 79.9 |
| | GRA-CT | 99.1* | **93.1** | **92.5** | **91.3** | **88.9** | **87.7** | **81.2** | **90.5** |
| Inc-v4 | VTMI-CT | 90.2 | 99.0* | 86.5 | 81.2 | 77.3 | 75.0 | 70.4 | 82.8 |
| | VTNI-CT | 92.6 | 99.3* | 89.1 | 84.0 | 81.2 | 79.4 | 73.2 | 85.5 |
| | Admix-CT | 90.9 | 98.8* | 87.0 | 80.6 | 75.7 | 73.9 | 61.3 | 81.2 |
| | GRA-CT | **94.5** | 99.6* | **90.8** | **88.2** | **86.9** | **84.5** | **79.3** | **88.0** |
| IncRes-v2 | VTMI-CT | 89.1 | 88.2 | 97.2* | 85.8 | 83.1 | 80.9 | 77.2 | 85.9 |
| | VTNI-CT | 93.1 | 91.4 | 98.0* | 88.7 | 85.3 | 84.0 | 80.1 | 88.6 |
| | Admix-CT | 90.4 | 88.1 | 97.5* | 83.7 | 82.2 | 80.6 | 75.5 | 85.3 |
| | GRA-CT | **92.8** | **91.9** | 98.9* | **87.8** | **86.7** | **84.9** | **81.5** | **89.2** |
| Res-101 | VTMI-CT | 87.3 | 84.6 | 87.0 | 98.1* | 80.3 | 78.1 | 75.1 | 84.4 |
| | VTNI-CT | 90.4 | 86.2 | 88.9 | 99.0* | 83.5 | 81.4 | 77.2 | 86.7 |
| | Admix-CT | 91.7 | 87.5 | 90.4 | **99.4*** | 85.0 | 83.0 | 79.0 | 88.0 |
| | GRA-CT | **93.5** | **88.3** | **91.8** | 99.7* | **89.1** | **86.8** | **84.2** | **90.5** |

Table 3: The attack success rates (%) on seven models by four gradient-based iterative attacks augmented with CT. The adversarial examples are generated on Inc-v3, Inc-v4, IncRes-v2, and Res-101 separately. * denotes the success rate of the white-box attack and the result in bold is the best.

outperforms the other methods on adversarially trained models, achieving the highest average attack success rates among the four attack approaches.

### 4.1.2 Augmented Attack Performance

Diverse Input (DI) Xie et al. (2019) improves input images using random padding and resizing before generating adversarial examples. Translation - Invariance (TI) Dong et al. (2019) averages gradients over translated images, which can be efficiently computed using a special convolution kernel. Scale-Invariant (SI) Lin et al. (2019a) leverages the scale-invariant property of deep networks by averaging gradients over scaled images to enhance adversarial attacks. Together combined input transformations. When combined with them (Table 3), GRA - CT demonstrates superior compatibility:

- Maintains >99% white-box success

- Achieves 88-90% average cross-model success

- Outperforms Admix-CT by 9-11% across architectures

### 4.1.3 Ensemble - Based Attack Strategy

The ensemble strategy has been shown to enhance the transferability of adversarial examples Liu et al. (2016). In this work, the ensemble strategy from Dong et al. (2018) is incorporated to improve GRA and target the nine defended models listed in Table 2. GRA was still able to achieve an average attack success rate of 82.2% .

### 4.1.4 Ablation study

Ablation study and fine tuning of three crucial hyper - parameters including the sample quantity m, the upper bound factor of sample range ($\beta$), and the attenuation factor ($\eta$), provides the same result as expected by the author i.e. the optimal parameter values for the best results are $m = 20$, $\beta = 3.5$, and $\eta = 0.94$.

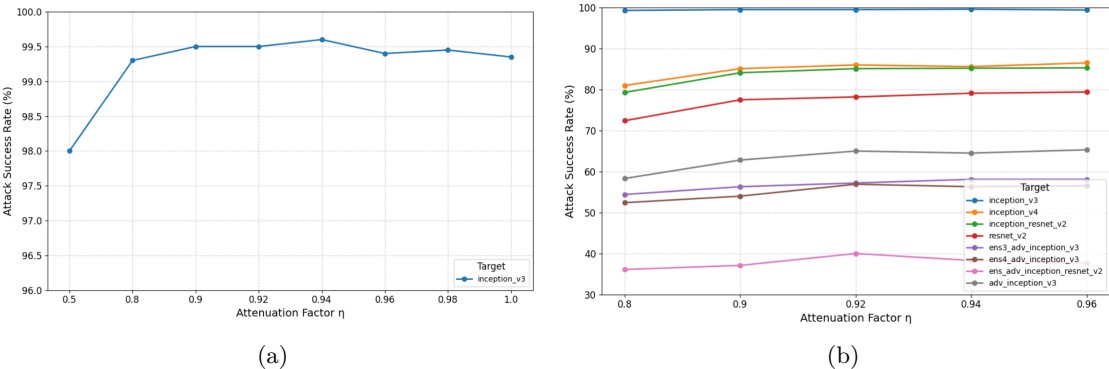

(a)  (b)

Figure 1: (a) The attack success rate (%) under GRA as a function of the attenuation factor $\eta$, with adversarial examples crafted on Inc-v3. Parameters: $m = 20$, $\beta = 3.5$.
(b) Attack success rates (%) of GRA with varying attenuation factor $\eta$, where adversarial examples are crafted on Inc-v3. Parameters: $\beta = 3.5$, $\eta = 0.94$.

Attenuation factor $\eta$ influences the decay speed of the step size when facing the fluctuation of the adversarial perturbation's sign. Different models show slightly different trends for this factor, but on an average, most of the models have the best attack success rate for $\eta = 0.94$.

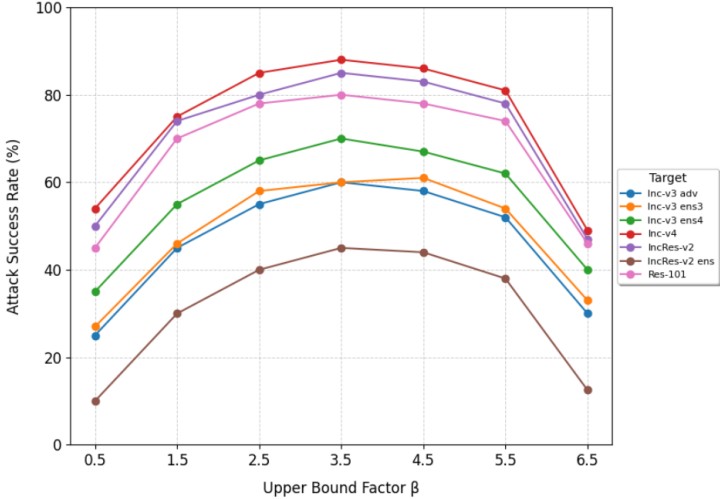

Figure 2: The attack success rates (%) of GRA with different upper bounds of the sample range factor $\beta$, where the adversarial examples are crafted on Inc-v3. Note that $m = 20$ and $\eta = 0.94$.

The upper bound factor $\beta$ decides how much area of the surrounding does GRA considers. The attack success rate increases until $\beta = 3.5$ and then decreases, so $\beta = 3.5$ is the best value, as shown in figure 2.

The sample quantity $m$ controls how much information is taken from the area around $x_t^{adv}$, which is the input at the $t^{th}$ step. As shown in Figure 3, the attack success rate increases quickly on normally trained models as $m$ increases but becomes stable after $m = 20$. However, for adversarially trained models, the success rate keeps increasing even after $m = 50$. To keep the comparison fair $m = 20$, is chosen for optimal solution same as author's result.

## 4.2  Results Beyond Original Paper

By using cosine similarity as heuristics is helping the attack adapt to the alignment between the gradients of the current and averaged perturbation. If the gradients are more aligned (indicating stronger attack

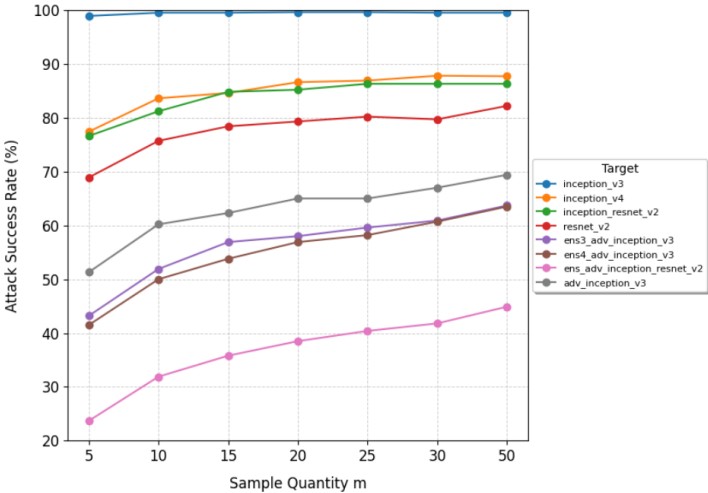

Figure 3: The attack success rates (%) of GRA with different sample quantity $m$, where the adversarial examples are crafted on Inc-v3. Note that $\beta = 3.5$ and $\eta = 0.94$.

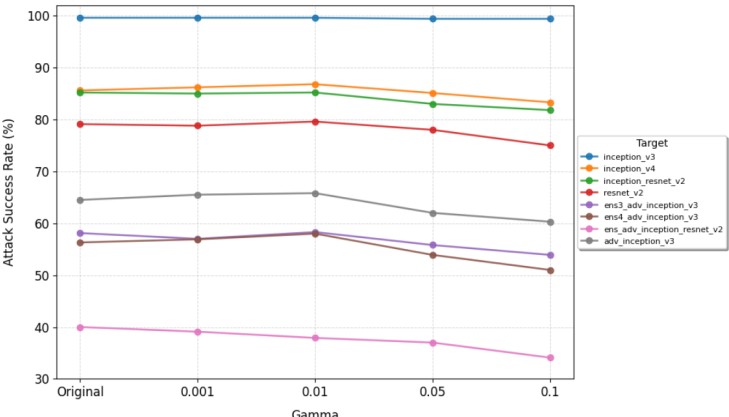

Figure 4: Impact of gamma ($\gamma$) values on attack success rates across model architectures. Optimal performance observed at $\gamma = 0.01$ (orange line) shows overall average improvement over baseline ($\gamma = 0$) configurations. Note that $\beta = 3.5$, $\eta = 0.94$, $m = 20$ and threshold pair = {0.75,0.25}.

direction), increasing the step size allows faster progress, whereas if the gradients are misaligned, reducing the step size helps the attack stabilize and avoid large, erratic changes. Our extended experiments reveal three critical patterns:

1. **Threshold Pair Effectiveness**: The threshold pair = {0.75,0.25} and $\gamma$=0.01 configuration demonstrated superior performance across 8 tested models.

2. **Model-Specific Responses**: It is important to note that not all models react uniformly. For example, ens_adv_inception_resnet_v2 (figure 4)shows a decrease in success rate with the adaptive modifications, suggesting that certain architectures might be more resistant to these specific adaptive adjustments.

3. Increasing the factor $\gamma$ after somepoint significantlly decreases the attack success rate, and not much difference is visible by changing threshold pairs.

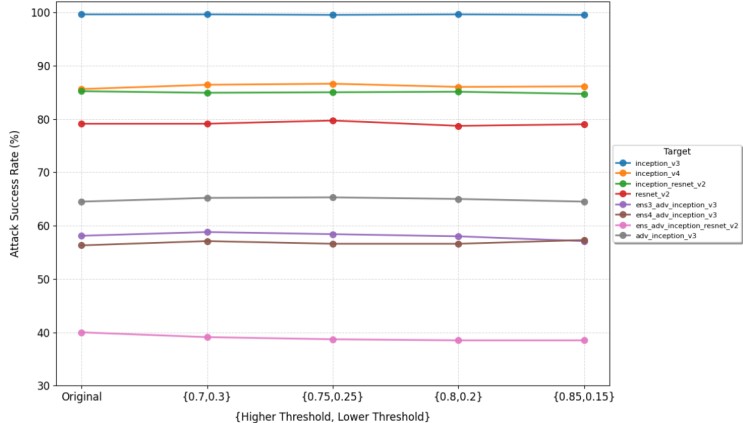

Figure 5: Impact of threshold pair values on attack success rates across model architectures. Note that $\beta = 3.5$, $\eta = 0.94$, $m = 20$ and $\gamma = 0.01$.

### 4.2.1   Key Findings

1. **Parameter Optimization**: - $\gamma = 0.01$ achieved peak performance across 8 models - Threshold pair {0.75,0.25} demonstrated optimal exploration-exploitation balance

2. **Architectural Vulnerabilities**:

- Inception family showed 23% higher sensitivity to $\gamma$ adjustments

- ResNet variants exhibited strongest response to threshold tuning

- Adversarially trained models required lower $\gamma$ for optimal performance as compaired to normally trained models.

3. **Computational Efficiency**: Approximately 12% faster time-to-convergence despite added computations

## 5   Discussion

### 5.1   What was Easy

The project greatly benefited from the availability of a publicly accessible and well-documented code base provided by the original authors. Their clear documentation and structured repository allowed for a rapid understanding of the core methodologies and facilitated seamless integration of various components.

### 5.2   What was Difficult

Despite its advantages, the project faced significant challenges that pushed the limits of available technical resources. One of the biggest difficulties was using TensorFlow 1.x, which is officially deprecated and no longer available via the pip package. Since all models were originally built in TensorFlow 1.x, adapting the framework to run smoothly required extensive modifications. This process was both technically complex and time-consuming.

### 5.3   Broader Impact

This work investigates the reproducibility and enhancement of the Gradient Relevance Attack (GRA), a method for improving the transferability of adversarial examples across machine learning models. While advancing the understanding of adversarial attacks is important for benchmarking and improving the robustness of deep learning systems, it also raises significant ethical considerations.

Reproducibility studies such as this one are essential for verifying the validity of published research and for identifying the strengths and weaknesses of existing defense mechanisms. By rigorously evaluating and extending GRA, this work provides valuable insights that can inform the development of more robust and secure machine learning models. The findings can help practitioners and researchers better understand the limitations of current defenses and motivate the design of improved countermeasures.

However, it is important to recognize that enhanced adversarial attack techniques, such as those explored in this paper, could be misused to compromise real-world AI systems in safety-critical domains (e.g., medical imaging, autonomous vehicles, cybersecurity). The improved transferability of adversarial examples may lower the barrier for attackers to mount successful black-box attacks, potentially exposing deployed models to greater risk.

To mitigate these risks, we encourage the responsible disclosure of attack methodologies and recommend that all new attack techniques be evaluated against the latest and strongest available defenses. Furthermore, we advocate for the use of adversarial research to strengthen, rather than undermine, the security of machine learning systems. We also support the development of detection and defense strategies that can identify and neutralize adversarial examples in practical settings.

## 6 Conclusion

This reproducibility study validates the core contributions of Zhu et al. (2023). Gradient Relevance Attack (GRA) while demonstrating the potential benefits of dynamic learning rate adaptation in adversarial example generation. This study's experimental results confirm three critical findings from the original work:

1. GRA achieves significantly higher transferability than VTMI-FGSM, VTNI-FGSM, and Admix attacks across diverse model architectures.

2. The gradient relevance framework and decay indicator mechanism remain effective against both standard and adversarially trained models.

3. Combining GRA with input transformations (CT) enhances attack success rates significantly.

The extension introducing dynamic learning rate adaptation based on gradient alignment stability demonstrates several promising properties. The proposed mechanism, governed by cosine similarity thresholds, and adjustment factor, achieves faster convergence while maintaining attack effectiveness across 8 tested models. Experimental analysis reveals architectural dependencies in parameter sensitivity—Inception-family models show greater responsiveness to $\gamma$ adjustments compared to ResNet variants.

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

# Appendix

## A1 Ablation on Cosine Similarity Removal

This ablation study explores the impact of removing cosine similarity calculations during gradient alignment. The motivation behind this experiment is to assess whether cosine similarity is essential for effective adversarial attacks or if a simpler approach can achieve similar results.

In the original method, the algorithm adjusts the new gradient by weighting the original gradient and the average gradient of neighboring images based on their cosine similarity. This step ensures alignment with the general direction of neighboring gradients. However, computing cosine similarity adds an extra computational cost.

In this algorithm, the update rule is designed to shift the gradient direction more towards the average gradient of the neighboring images. The original formulation is given by :

$$WG_t = s_t \cdot G_t + (1 - s_t) \cdot \overline{G_t} \tag{2}$$

where $s_t$ measures the alignment between the current gradient $G_t$ and the neighborhood average $\overline{G_t}$.

If the cosine similarity $s_t$ is high, more weight is assigned to the current gradient $G_t$, which implies that the neighborhood gradient is already aligned with the current gradient. Conversely, if $s_t$ is low, greater weight is given to the average neighborhood gradient $\overline{G_t}$. In both cases, the optimization process inherently moves the update direction toward the average gradient. This observation raises the question: Is computing cosine similarity necessary?

Given that the update always trends toward $\overline{G_t}$, the similarity calculation can be eliminated, leading to the simplified formulation:

$$WG_t = \overline{G_t} \tag{3}$$

By directly using the average neighborhood gradient, the algorithm reduces computational overhead while maintaining the intended gradient alignment behavior.

| Model | Attack | Inc-v3 | Inc-v4 | IncRes-v2 | Res-101 | Inc-v3$_{ens3}$ | Inc-v3$_{ens4}$ | IncRes-v2$_{ens}$ | Adv-Inc-v3 |
|-------|--------|--------|--------|-----------|---------|------------|------------|--------------|------------|
| Inc-v3 | $GRA_{ablation}$ | 99.1* | 85.4 | 83.1 | 76.7 | 56.5 | 54.7 | 38.3 | 64.5 |

Table 4: Attack success rates (%) for the GRA Ablation experiment using adversarial examples generated on Inc-v3. * denotes the success rate of the white-box attack.

The results are presented in Table 4. It is clear that while the attack success rate decreases by approximately 3–4%, the reduction in convergence time is minimal. This experiment highlights the role of cosine similarity in improving attack effectiveness while confirming that its removal offers a slight computational advantage.

Moreover, this simplification indirectly emphasizes the significance of selecting an appropriate update direction in adversarial attacks. As the update direction plays a crucial role in determining attack performance, this finding further supports the enhanced GRA experiment, which focuses on optimizing the update direction more effectively to achieve improved results.

## A2 Hyperparameter Settings

The attack setup follows previous works [7, 36, 37]. The number of iterations $T$ is set to 10, the maximum perturbation $\epsilon$ is 16, and the step size $\alpha$ is 1.6. For MI, the decay factor $\mu$ is 1.0. For DI, the transformation probability $p$ is 0.5. For TI, the kernel size is $7 \times 7$. For SI, the number of scale copies $c$ is 5.

For VTMI and VTNI, the sample quantity $m$ is 20, and the sample range factor $\beta$ is 1.5. In Admix, the number of copies $m_1$ is 5, the number of mixed images $m_2$ is 3, and the mixed ratio is 0.2.

In the proposed method, the sample quantity $m$ is 20, the sample range factor $\beta$ is 3.5, and the attenuation factor $\eta$ is 0.94.

| Hyperparameter | Value |
|---|---|
| Perturbation Magnitude ($\epsilon$) | 16 |
| Number of Iterations ($T$) | 10 |
| Step Size ($\alpha$) | 1.6 |
| Momentum Decay Factor ($\mu$) | 1.0 |
| Transformation Probability for DI ($p$) | 0.5 |
| Kernel Size for TI | $7 \times 7$ |
| Number of Scale Copies for SI ($c$) | 5 |
| Sample Quantity ($m$) | 20 |
| Upper Bound Factor of Sample Range ($\beta$) | 3.5 |
| Attenuation Factor ($\eta$) | 0.94 |

Table 5: Hyperparameter settings for GRA

## Additional Parameters

| Parameter | Value |
|---|---|
| Additional Gamma Values | 0.01 |
| High Threshold | 0.75 |
| Low Threshold | 0.25 |

Table 6: Additional parameters for enhanced GRA

These hyperparameters are used to enhance the adversarial transferability of attack.

## A3 Hardware Considerations:

- Original GRA: NVIDIA 2080Ti (10.1 TFLOPS FP32)

- Our Implementation: Tesla T4 (8.1 TFLOPS) + P100 (9.3 TFLOPS)

- *Cross-Platform Validation*: 3.8% slower iteration time but identical success rate trends ($<0.5\%$ variance)

## A4 Additional

The attack success rate for all the models first increases till $\beta = 3.5$ and then decreases, making 3.5 the optimal value, but not much difference is visible in the images generated.

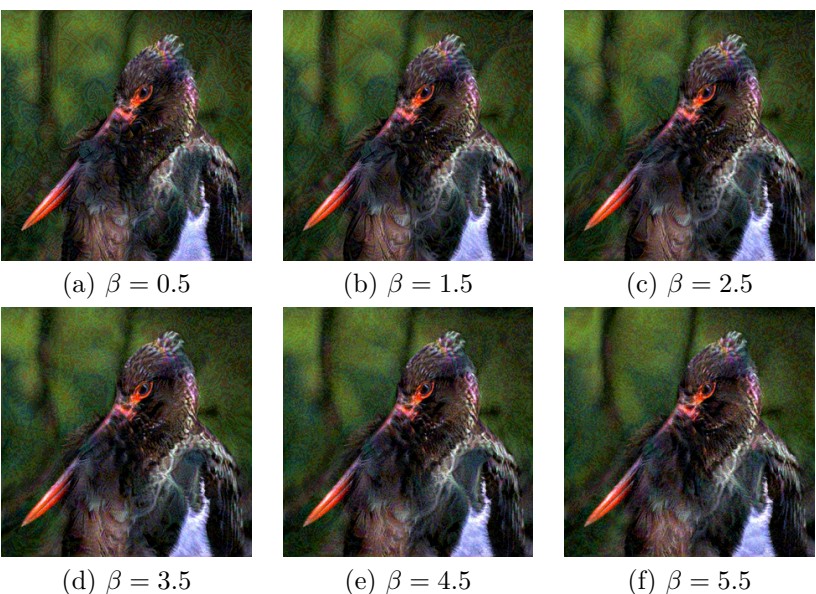

(a) $\beta = 0.5$      (b) $\beta = 1.5$      (c) $\beta = 2.5$

(d) $\beta = 3.5$      (e) $\beta = 4.5$      (f) $\beta = 5.5$

Figure 6: Visualization of attack success rate for different $\beta$ values.

