# OpenReview forum: "Reproducibilty Study of Boosting Adversarial Transferability via Gradient Relevance Attack"
_TMLR — Rejected by TMLR_

### Review · Reviewer_mkQs · 2025-03-21

**Summary Of Contributions:**

The paper reproduced the results of the paper proposing the Gradient Relevance Attack (GRA) method and found that the achieved attack
success rates were within a $1\%$ margin of those reported in the original study, confirming the effectiveness of the GRA method. Then, the paper proposes an extension of GRA, which introduces a dynamic learning rate $\alpha$. The proposed method achieves comparable performance compared to the original GRA.

**Audience:**

No

**Broader Impact Concerns:**

There is no Broader Impact Statement section in this paper.

**Claims And Evidence:**

Yes

**Requested Changes:**

I hope all the writing issues listed in the **Weaknesses** part can be solved.

**Strengths And Weaknesses:**

- The contribution of this paper is not large enough to be published. The two contributions of this paper are: (1) reproductive results for GRA; (2) an extension of GRA. However, the effectiveness of GRA is already shown in the original paper that proposed GRA. Moreover, the extension is based on a simple and intuitive assumption that when the gradients between a perturbation set align well, the direction is very likely to increase the loss. However, the performance of the proposed method is only comparable to GRA, which means that the good performance of the proposed methods is not from the step size adjustment strategy but the GRA method itself.
- The writing is not so good and some facts are not stated clearly. For example:
  - In equation (2), $M_ t, M_ t^e, M_ t^d$ are used. However, the authors do not show the initial value $M_1$, which makes the iterative equations confusing. Moreover, the definition of $M_ t^e$ and $M_ t^d$ are not given, there is only a sentence that "$M_ t^e$ is 1 if the accumulated gradients between the iterations have the same sign and $M_ t^d$ is 1 if they have a different sign". Such informal descriptions may cause misunderstanding. To make the readers understand GRA better, $M_ t, M_ t^e, M_ t^d$ should be formally defined.
  - On line 6 of Algorithm 1, the notation $\mathcal{U}$ is not defined, what distribution does it stand for? After reading the original paper of GRA, I found that $\mathcal{U}$ stands for the uniform distribution. Moreover, the $d$-dimensional uniform distribution with upper bound $\beta \epsilon$ and lower bound $-\beta \epsilon$ should be written as $\mathcal{U}\left( \left[  -\beta \epsilon,  \beta \epsilon \right]^{d} \right)$, the expression $\mathcal{U}\left( -(\beta \epsilon)^d, (\beta \epsilon)^d \right)$ is confusing and makes people think it is a one-dimensional uniform distribution over the interval $\left[  -(\beta \epsilon)^d, (\beta \epsilon)^d \right]$.
  - In the paragraph after Algorithm 1, the paper says "At each iteration, small random noise, sampled from a Gaussian distribution", however, according to the original paper of GRA, the random noise is sampled from a uniform distribution but not Gaussian distribution. I wonder what is the noise distribution used in the reproductivity experiments.
  - In equation (3), the subscripts of $\bar{s}_ t, \tau_ {high}, \tau_ {low}$ are not used correctly.
  - On line 7 of Algorithm 2, what does $s_t^i$ stand for? The paper does not make it clear. Moreover, if $s_t^i$ is the cosine similarity between $G_ t(x)$ and $G_ t(x_ i)$, then $\mathbb{E}[s_ t] \ne \frac{1}{m} \sum_ {i=1}^m s_ t^i$ since $\frac{1}{m} \sum_ {i=1}^m s_ t^i$ is random due to the randomness of the choice of the noises.
  - In the caption of Table 1, the paper says "the result in bold is the best", however, there is no bold result in the table.

---

> ### Author Response · Authors · 2025-04-20
>
> We are grateful for the review. We have now addressed your concerns:
>
> - Minor contribution and intuitive extension: While our extension is based on an intuitive assumption, we show that it accelerates convergence and performs comparably under different conditions, offering practical value, a relevant extension for the reproducibility study. We now highlight these findings more explicitly.
>
> - Clarity and writing issues: We thoroughly revised all confusing equations and informal descriptions.  We also reorganized the paper to improve logical flow, and the overall structure.
>
> - Missing broader impact: We added a dedicated Broader Impact section, discussing both potential risks of misuse and the importance of reproducibility for defense development.
>
> We hope this addresses your concerns.

---

### Review · Reviewer_fX3B · 2025-03-31

**Summary Of Contributions:**

This reproducibility study validates the effectiveness of the Gradient Relevance Attack (GRA) method in enhancing adversarial transferability across diverse models, achieving attack success rates within 1% of the original results. The study extends GRA by introducing a dynamic learning rate mechanism that adjusts step size based on cosine similarity between gradients, accelerating convergence and improving attack performance in certain scenarios. Additionally, the work provides a detailed analysis of hyperparameter tuning and demonstrates GRA's robustness against both standard and adversarially trained models.

**Audience:**

Yes

**Broader Impact Concerns:**

The paper should include a Broader Impact Statement discussing the implications of enhancing adversarial attack transferability. Specifically, the work could be misused to develop stronger attacks, posing security risks to machine learning systems. A discussion on potential safeguards and ethical considerations is recommended.

**Claims And Evidence:**

Yes

**Requested Changes:**

**Critical Changes:**

1. Clarify the motivation for reproducing the GRA. The paper should explicitly state why reproducing GRA is necessary, what insights it provides, and how it contributes to the field.
2. Improve the overall structure of the paper to enhance readability and logical flow. Consider reorganizing sections to make the contributions and experimental findings clearer.
3. Update the comparison baselines in the experiments. The paper should include more recent and relevant methods to ensure a fair and meaningful evaluation.
4. Ensure consistency in hardware setup. The reproduction experiment should use hardware configurations as close as possible to those in the original GRA study to maintain rigor and comparability. If hardware differences are unavoidable, justify their impact on the results and provide necessary adjustments in experimental settings.
5. Define all notations used in algorithms and equations. Unspecified symbols can lead to ambiguity and hinder reproducibility.

**Suggested Improvements:**

1. Enhance the explanation of the quantization experiment. Clearly describe how the quantization experiment demonstrates the improvements of the extended GRA over the original GRA. Provide quantitative results and analysis to support the claims.
2. Strengthen the contributions section. Highlight key findings and unique aspects of the work to better distinguish it from prior research.

**Strengths And Weaknesses:**

**Strength：**

1. Successfully reproduces the GRA and validates its experimental results.
2. Introduces a dynamic learning rate adjustment mechanism, which enhances model convergence speed.

**Weakness:**

1. The motivation for reproducing the GRA is not clearly stated.
2. The overall structure of the paper is not well-organized.
3. The choice of comparison baselines in the experiments is outdated.
4. The contributions of the paper are not particularly significant.
5. The hardware setup used in the paper (Tesla T4 and P100) differs from that of GRA (Nvidia 2080Ti and Tesla V100), which undermines the rigor of the reproduction experiment.
6. Some notations in the algorithm and equations are not fully defined.
7. The quantization experiment for evaluating the improvements of the extended GRA over the original GRA is not clearly explained.

---

> ### Author Response · Authors · 2025-04-20
>
> We deeply appreciate the kind review. We have addressed the changes as requested.
> -  Unclear motivation and structure: We have added the motivation section to clearly outline the reproducibility gaps and potential gains from dynamic adaptation. We also reorganized the paper to improve logical flow and the overall structure.
>
> - Hardware discrepancy: We added a discussion on hardware discrepancies in the appendix. We justify how differences (e.g., GPU types) do not practically affect our conclusions.
>
> - Undefined notations and algorithm details: All notations and algorithmic steps have been revised for clarity.
>
> - Broader Impact Statement missing: We added a dedicated Broader Impact section, discussing both potential risks of misuse and the importance of reproducibility for defense development.
>
> We hope this addresses all your concerns.

---

### Review · Reviewer_qH94 · 2025-04-07

**Summary Of Contributions:**

This paper presents a reproducibility study of the Gradient Relevance Attack (GRA), which aims to enhance the transferability of adversarial examples. The authors successfully replicated the core findings and  extended the study by introducing and evaluating a novel dynamic learning rate mechanism.

**Audience:**

No

**Broader Impact Concerns:**

This work reproduce a transfer attack method, which may be utilized by malicous users to attack DNN models.

**Claims And Evidence:**

Yes

**Requested Changes:**

See the weakness.

**Strengths And Weaknesses:**

### Strength

- This paper reproduces GRA successfully with codes.
- This paper extended the original GRA method.

### Weakness

- Contribution is limited.  Its main content primarily reproduces the same results as GRA , without introducing any particularly insightful findings. Though some experimental techniques are presented, they do not significantly enhance the overall contribution. This work would be more suitable for a workshop or a blog rather than TMLR.
- The importance of GRA should be clearly articulated. Why was GRA specifically chosen for reproduction?
- The paper emphasizes the challenges of transfer, which are addressed by GRA. What’s your conctribution on these challenges?

---

> ### Author Response · Authors · 2025-04-20
>
> We appreciate the concerns raised by the reviewer and have tried to address them to the best possible:
>
> - We acknowledge the concern and have strengthened the motivation and positioning of our work. In particular, we now clearly state why GRA was chosen—due to its influence in transfer-based attacks and its underexplored mechanisms. We emphasize the value of reproducibility, especially in adversarial machine learning, and demonstrate that our extension (dynamic learning rate) offers concrete practical benefits.
>
> - We now better articulate the relevance of GRA and explain how our dynamic learning rate mechanism tackles transfer-related convergence challenges.
>
> We hope this addressed all your concerns.

---

### Decision · Action_Editor_v5iV · 2025-05-02

**Recommendation:** Reject

**Comment:**

All the reviewers indicate some issues on the limited contributions, the experimental setup, the fairness of the comparison and the clearness of the writing. Many notations in the algorithm and equations are not well defined, which lead to ambiguity and hinder reproducibility. While the authors provide responses to the reviewers comments, all the reviewers are not quite satisfied with their replies. In particular, the authors do not introduce strong baselines to support or enhance the empirical evaluation.

**Audience:**

GRA is a useful method to improve the transferability of adversarial examples and black-box adversarial attacks. The paper presents a reproducibility study of the GRA, which should be interesting to some TMLR audiences.

**Claims And Evidence:**

The paper presents a reproducibility study of the Gradient Relevance Attack (GRA) and confirms its effectiveness. The paper also proposes an extension of GPA by introducing a dynamic learning rate, and shows that it achieves a comparable performance as GRA. All reviewers agree that a reproducibility study of GRA is meaningful. Meanwhile. all reviewers have concerns about the contributions of the paper. For example, the baseline methods used in the paper are outdated and the authors need to use more recent baseline methods for a fair comparison. Also, the hardware setting is different from the hardware configuration in the original GRA study, which may lead to misleading results. The writing is also not satisfactory as there are many inconsistent and not well-defined notations.